# Resource Prediction-Based Edge Collaboration Scheme for Improving QoE

**DOI:** 10.3390/s21248500

**Published:** 2021-12-20

**Authors:** Jinho Park, Kwangsue Chung

**Affiliations:** Department of Electronics and Communications Engineering, Kwangwoon University, Seoul 01897, Korea; jhpark@cclab.kw.ac.kr

**Keywords:** Internet of Things (IoT), edge computing, mobile edge computing (MEC), computation offloading

## Abstract

Recent years have witnessed a growth in the Internet of Things (IoT) applications and devices; however, these devices are unable to meet the increased computational resource needs of the applications they host. Edge servers can provide sufficient computing resources. However, when the number of connected devices is large, the task processing efficiency decreases due to limited computing resources. Therefore, an edge collaboration scheme that utilizes other computing nodes to increase the efficiency of task processing and improve the quality of experience (QoE) was proposed. However, existing edge server collaboration schemes have low QoE because they do not consider other edge servers’ computing resources or communication time. In this paper, we propose a resource prediction-based edge collaboration scheme for improving QoE. We estimate computing resource usage based on the tasks received from the devices. According to the predicted computing resources, the edge server probabilistically collaborates with other edge servers. The proposed scheme is based on the delay model, and uses the greedy algorithm. It allocates computing resources to the task considering the computation and buffering time. Experimental results show that the proposed scheme achieves a high QoE compared with existing schemes because of the high success rate and low completion time.

## 1. Introduction

Smart Internet of Things (IoT) devices are becoming increasingly popular and play an increasingly important role in every aspect of our daily lives [1]. IoT-based applications require different deadlines, bandwidths, and high computation performance [2]. However, IoT devices do not have enough computing resources to satisfy the high computation performance. This resource requirement is filled by remote and cloud centers or cloud services [3]. However, a cloud center is located at a long distance from the device and cannot satisfy recent time-sensitive IoT applications, such as augmented reality (AR), virtual reality (VR), and video analysis. In addition, the backhaul network is congested by tasks sent by a large number of devices to the cloud [4]. Edge servers have been proposed to solve these challenges.

Recently, a new computing paradigm edge server, called mobile edge computing (MEC), was proposed by the European Telecommunications Standards Institute (ETSI) [5]. These edge servers—also called, “the cloud at the edge of the network”—are attracting attention as an important component in 5G network technology. Edge servers process applications in real-time according to low communication latency since they are located close to the device. In addition, an edge server is suitable for computation-intensive tasks due to sufficient computation resources compared with the device [6]. Nevertheless, edge servers have a problem in that processing efficiency degrades as several devices are connected to the edge server. An edge collaboration scheme was proposed to solve this.

The existing edge collaboration scheme determines the collaboration target without considering the computing resource or network resource [6,7]. Therefore, the task completion time is increased by cooperating with an edge server with a higher load even though there is an edge server with a relatively lower load. The edge collaborations increase the task completion time according to computation time, thereby lowering the quality of experience (QoE). When network resource is not considered, the edge server selects the collaboration target based on the task processing time. The collaboration scheme cooperates with an edge server far away from the local edge server, even though a nearby edge server is appropriate. Thus, QoE is degraded due to increased completion time and decreased success rate as a result of the high communication time.

Edge server processes allocate computing resources to assigned tasks by a collaboration algorithm. In the existing allocation scheme, tasks assigned to the edge servers share computing resources. When the number of processing tasks in the edge server is small, sufficient computing resources are allocated, and the completion time to meet the deadline is low. However, as the number of processing tasks increases, the allocated computing resources decrease, and QoE is affected.

QoE is affected by the collaboration target determined by the edge server. There are two main QoE factors: completion time and success rate of the tasks. The collaborative target needs a minimum amount of computing resources to meet task requirements and improve the task’s success rate. If the edge collaboration scheme only considers the success rate of a task, the completion time of a task is high due to the collaboration with the high-loaded edge server. When the edge collaboration scheme only considers the completion time of the task, the edge server selects a collaboration target considering only the average completion time of tasks. Therefore, even if the average completion time of tasks is the same, an edge server that can improve the average success rate is not selected as a collaboration target.

In this paper, we propose a resource prediction-based edge collaboration scheme for improving QoE. The proposed scheme probabilistically determines collaboration based on computing resource prediction when the edge server receives a task. According to the collaboration decision, an optimal collaboration target is selected to improve the success rate of the task. We first predict the completion time of the task to select the collaboration target. Completion time is calculated using the processing time, communication time, and buffering time. Second, we formulate the edge collaboration problem. To solve the problem, we use the greedy algorithm based on the predicted completion time. To guarantee the QoE of tasks assigned according to the above algorithm, we allocate computing resources. Computing resource allocation is determined based on the number of tasks to be processed by the edge server according to the trade-off between computation and buffering time.

The remainder of this paper is organized as follows: Section 2 describes the evolution of computation offloading and existing collaboration schemes. Section 3 presents collaboration target decision and computing resource allocation for reducing the completion time and improving the success rate. In Section 4, the experimental results for the proposed scheme are presented. Finally, the conclusions are presented in Section 5.

## 2. Related Work

In this section, we describe the computation offloading evolution and existing collaboration schemes.

### 2.1. Computation Offloading Evolution

Computation offloading is aimed at improving application performance using a powerful infrastructure. This paradigm has evolved over the recent years with the development of various infrastructures.

In the 1990s, *Odyssey* [8], which considers application behaviors such as CPU, bandwidth, and battery power dynamically, was proposed to improve mobile application performance and save battery life.

In 2001, M. Satranaraynan [9] proposed a computation offloading concept called “cyber foraging” that augments computing capability through wired computation nodes.

In the 2000s, cloud center and mobile computing began gaining attention in industry and real life. Amazon released the elastic compute cloud (EC2) in 2006 [10]. With the development of various mobile operating systems, various applications are being developed that are useful in our daily lives. Consequently, technologies that combine mobile and cloud computing, such as mobile cloud computing (MCC) [11], have been proposed. MCC has become a de-facto standard for computation offloading and has been involved in the research of several schemes such as *MAUI* [12], *Cuckoo* [13], *CloneCloud* [14], *ThinkAir* [15], and *COMET* [16].

However, with increasing real-time applications, such as AR, video analytics, and VR, computation offloading to the cloud cannot guarantee task requirements. Therefore, researchers have focused on using computing devices located nearby to reduce network delay. M. Satyanarayanan et al. [17] proposed “cloudlet” as a computing node between the devices and the cloud. In 2012, Cisco announced the role of fog computing in 2012 [18], and MEC was proposed by ETSI [5].

### 2.2. Existing Collaboration Scheme

As a popular optimization method in distributed computing systems, collaboration schemes, such as cloud computing, have been intensively studied in recent years [19]. J. Liu et al. [20] used the Markov chain theory to analyze the queuing delay in each waiting task and estimate the power consumption of an IoT device. H. Tan et al. [21] analyzed task processing in the real world. Because stochastic optimization requires a pre-known distribution, an optimization algorithm that can minimize the weighted completion time with power constraints was designed.

Other collaboration schemes consider the delay in the computation time. *CACTA* [22] considers the uncertainty and stochastic nature of the edge and formulates an optimization problem by modeling the computation capacities and costs of edge nodes using the autoregressive integrated moving average (ARIMA) model. This scheme improves the transmission efficiency of tasks and reduces the task completion time. Y. Zhang et al. [23] formulated the utilization cost of computation resources by determining the collaboration target using wholesale and buyback models to improve the quality of service (QoS). *LCDA* [24] classifies the task into latency-tolerant and latency-sensitive according to the processing deadline to select the collaboration target. By classified tasks, *LCDA* reduces task completion time and increases task success rates. When an edge server is overloaded, its performance tends to be low, which is evident by a high transmission time. H. Zhao et al. [25] selected edge servers or cloud centers using computation resources and task caching for improving processing and transmission efficiency. In contrast, *HOM* [26] proposed a heuristic method for communication delay minimization. This scheme uses a tree to find the shortest path to the destination edge server and set up a multi-hop network for performance evaluation. However, processing time is high because the load of the collaboration target is not considered. A. Jonathan et al. [27] proposed an edge collaboration scheme to consider edge locality. They monitored the neighboring node and selected the node based on network delay to satisfy the low latency of tasks.

Time delay and energy consumption are jointly studied in the literature. Zhuang et al. [28] assumed that the tasks cloud be divided into various sizes and formulated a mixed-integer nonlinear program to consider delay and energy consumption simultaneously. X. Chen et al. [29] researched multi-user cases with multi-channel wireless contentions.

Later, they jointly considered task offloading to share computation and communication resources for the multi-user scenario with multi-tasks [30]. J. Ren et al. [31] jointly selected edge cloud and edge server for minimizing latency by processing a task according to computing and communication resources. *HETO* [32] determines collaboration targets using weighted MAX-2SAT theory. In particular, *HETO* considers the asymmetry of communication cost and the heterogeneous computing resource of the server. *MSGA* [33] models applications based on directed acyclic graph (DAG). According to the application model, *MSGA* schedules tasks among edge servers. X. Gong [34] established the optimal communication scheduling and computation allocation. Based on communication scheduling and computation allocation, Gong found optimal computation nodes to reduce the task completion time. *D3PG* [35] partially offloads the task to an edge server. To minimize the task completion time, *D3PG* decides the offloading size of the task and network resource to the task based on reinforcement learning, which is designed based on a dueling and double network architecture to reduce the convergence time. S. Wang et al. [36] analyzed the convergence rate of the distributed gradient descent. Based on the analyzed results, they determined the best parameter between local update and global aggregation. *OL4EL* [37] supports both synchronous and asynchronous learning patterns, and can be used for both supervised and unsupervised learning tasks.

Existing schemes consider processing and transmission time in order to task completion time and process the task within the deadline. In this paper, we consider the processing and transmission time as well as the buffering time. In addition, we propose a scheme for allocating computing resources to process tasks, which reduces task completion time and processes the tasks within the deadline.

## 3. Proposed Scheme

In this paper, we propose an edge collaboration and computing resource allocation algorithm for improving the QoE in terms of task completion time and task success rate. The edge collaboration is designed by task completion time and a greedy algorithm, and the computing resource allocation is designed by processing and buffering time. In this section, we first introduce the edge collaborative network framework. Second, we focus on the decision-making process in the framework. Finally, the computing resource allocation scheme is described.

### 3.1. Edge Collaborative Network Environment

Figure 1 shows the edge collaborative network environment. The device is connected to the base station via a wireless network, and a large number of devices can be connected to the edge server. The edge server is located near the base station and monitors the base station and connected devices. The network between the edge servers is wired, and the edge servers collaborate for task processing. For edge collaboration, the edge server shares the tasks and resource information with other edge servers. In this paper, we make three major assumptions. First, the device does not have sufficient computing resources to process the task. Therefore, whenever a task is created, the device transmits the task and task information to a connected edge server. Second, the edge server knows the bandwidth of the network for task transmission. Finally, multiple tasks can be created concurrently using different devices. We define task information using Equation (1).
(1)$$W_{j}=\left\{L_{j},D_{j}\right\}$$

j denotes the index of the task, $L_{j}$ denotes the size of the task j, and $D_{j}$ denotes the processing deadline of the task j. The edge server processes the task received from the device and is aware of the task being processed $\left(E_{ij}=\left\{P_{1},P_{2},\cdots ,P_{J}\right\}\right)$. $E_{ij}$ denotes a set of information about tasks being processed in the edge server i. i denotes the index of the edge server. The task information recognized by the edge server is given by Equation (2).
(2)$$P_{j}=\left\{L_{j}^{r},R_{j}\right\}$$

$L_{j}^{r}$ denotes the remaining size until the task being processed by the edge server is completed and $R_{j}$ denotes the allocated computing resource for the task. Figure 2 shows the operation of the proposed scheme. $C_{i}$ denotes the computing resource usage, Th denotes the threshold of computing resource usage, and $C_{i}^{max}$ denotes the computing resource capacity of the edge server. The task created in the device is transmitted to the connected edge server using the wireless network. The edge server immediately decides offloading whenever it receives a task from the device. The computing resource usage of the edge server is predicated upon receiving the task from the devices. If the predicted computing resource usage is lower than the threshold, the task is processed in a local edge server; otherwise, a collaboration is determined probabilistically based on the predicted computing resource usage and threshold. We first model the completion time of the task and formulate an objective function to determine the optimal collaboration. To solve the complex problem of the objective function, a collaboration target is selected using a greedy algorithm. The tasks are then assigned to the edge server according to the selected collaboration target. If the edge server has available computing resources, the assigned tasks are processed immediately; otherwise, the task is stored in the edge server’s buffer. The edge server counts the number of offloaded tasks to be processed. The computing resource allocation is calculated by a trade-off between processing and buffering time.

**Figure 2 sensors-21-08500-f002:**
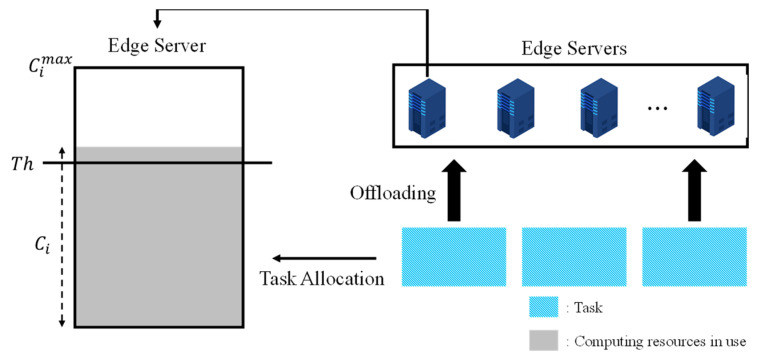
Operation of the proposed scheme.

### 3.2. Edge Collaboration

Figure 3 shows the flow chart of the collaboration decision in the proposed scheme. When the edge server receives a task from the device, the server calculates the computing resources required to add the processing of incoming tasks to those already being processed and the task received. The proposed scheme measures the load of the edge server based on the predicted computing resource usage; if it is lower than the threshold $\left(C_{i}^{max}\cdot Th\right)$, the edge server can provide enough computing resources for the task, and the task is then stored in the buffer of the local edge server. In the opposite case, the proposed scheme calculates the probability of task offloading (p) to reduce the load on the edge server and improve the utilization of computing resources. Task offloading is determined by comparing calculated probabilities with a randomly-generated value (v). The random value is generated between zero and one based on the Gaussian distribution. When v is lower than p, the task is processed by the local edge server to improve the utilization of computing resources. Conversely, if v is larger than p, the edge server finds the optimal collaboration target to reduce the load on the edge server. According to the received task, $C_{i}^{pred}$ of the edge server is predicted using Equation (3).
(3)$$C_{i}^{pred}=C_{i}+\frac{L_{\hat{j}}}{D_{\hat{j}}}$$

$\hat{j}$ denotes the index of the task received from the device and $L_{\hat{j}}$ denotes the size of the task $\hat{j}$. The probabilistic collaboration decision is then calculated using Equation (4).
(4)$$p=\underset{j}{\min}\left(\sqrt{\frac{C_{i}^{pred}}{C_{i}^{max}}\cdot \frac{E_{ij}}{D_{j}}}\right)$$
(5)$$s.t.\;C_{i}^{pred}>Th\cdot C_{i}^{max}$$

The decision to process the task locally versus collaborative offloading is determined according to the predicted computing resources and the remaining size of the task being processed by the edge server. As computing resource usage is expected to increase, we reduce the number of tasks assigned to edge servers to improve QoE. However, the load on the edge servers is reduced as the tasks get completed. Therefore, the proposed scheme reduces collaboration probability based on the state of the task being processed in the edge server.

**Figure 3 sensors-21-08500-f003:**
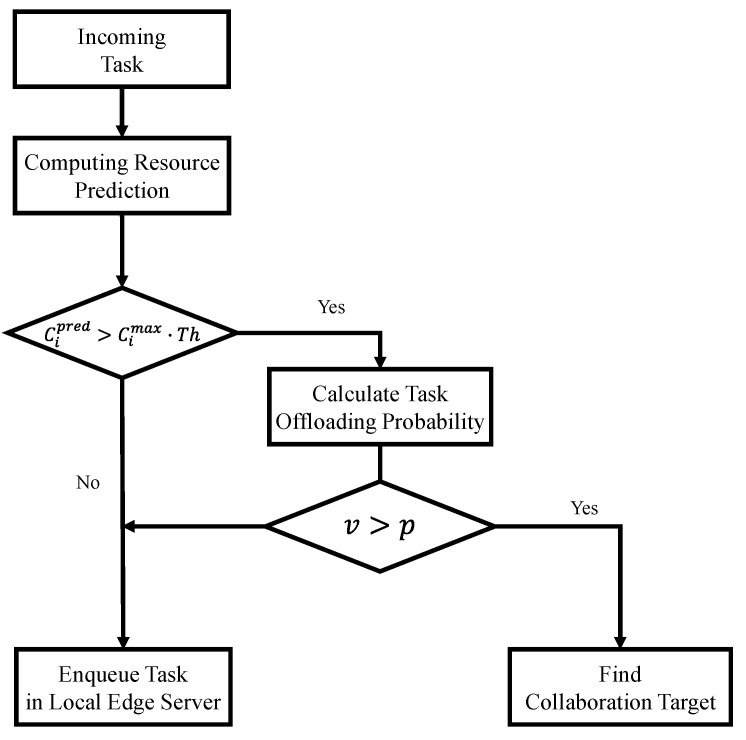
Flow chart of collaboration decision in the proposed scheme.

According to the probabilistic collaboration, the proposed scheme predicts the completion time of the task based on the delay model for selecting the collaboration target. The completion time of the task is determined based on the processing, communication, and buffering times. The processing time of the task $\left(T_{comp,\hat{j}}\right)$ is calculated according to the available computing resources of the edge server using Equation (6).
(6)$$T_{comp,\hat{j}}=\{\begin{array}{c} \frac{L_{\hat{j}}}{C_{i}^{avail}},\;if\;C_{i}^{avail}>0\\ \frac{L_{\hat{j}}}{E\left\{R_{j}\right\}},\;else \end{array}$$

$C_{i}^{avail}$ denotes the available computing resources in the edge server and $E\left\{R_{j}\right\}$ denotes the expectation value of $R_{j}$, which is calculated using the average input value. If the edge server does not have available computing resources, the task is processed after the task being processed on the edge server is complete. Therefore, the computation time of the task is calculated based on the computing resources allocated to the task being processed by the edge server. Communication time $\left(T_{comm,\hat{j}}\right),$ which is when an edge server collaborates with other edge servers, is calculated using Equation (7).
(7)$$T_{comm,\hat{j}}=\{\begin{array}{c} \frac{L_{\hat{j}}}{ABW_{i,i^{\prime }}},\;if\;i\neq i^{\prime }\\ 0,\;else \end{array}$$

$ABW_{i,i^{\prime }}$ denotes the bandwidth between the edge servers. If the edge server calculates itself as the only valid collaboration target, the task is immediately stored in the buffer; therefore, there is no transmission time. The buffering time of task $\left(T_{buf,\hat{j}}\right)$ is calculated using Equation (8).
(8)$$T_{buf,\hat{j}}=\{\begin{array}{c} a\left(E_{i}\right),\;if\;C_{i}^{avail}\leq 0\\ 0,\;else \end{array}$$

$a\left(E_{i}\right)$ denotes the delay function based on the task being processed in the edge server. The edge server immediately processes the task if it has available computing resources. Hence, the buffering time is zero. However, if the edge server does not have available computing resources, tasks are stored in the buffer and $a\left(E_{i}\right)$ is calculated using Equation (9).
(9)$$a\left(E_{i}\right)=\{\begin{array}{c} \overset{J}{\underset{j=1}{\sum }}\frac{L_{j}^{r}}{R_{j}}+\overset{J^{\prime }-J}{\underset{J^{\prime }=1}{\sum }}\frac{L_{j^{\prime }}}{R_{j^{\prime }}},\;if\;J^{\prime }>J\\ \overset{J^{\prime }}{\underset{j=1}{\sum }}\frac{L_{j}^{r}}{R_{j}},\;else \end{array}$$

J denotes the number of tasks being processed by the edge server and $J^{\prime }$ denotes the number of tasks stored in the buffer. The buffering time is determined by the number of tasks being processed in the edge server and stored in the buffer. If the number of tasks being processed is more than the number of tasks stored in the buffer, buffering time is predicted by the tasks being processed. When the number of tasks being processed is less than the number of stored tasks in the buffer, the task state does not change immediately to the processing state. The state of the task is changed after the tasks previously stored in the buffer are processed. Therefore, the buffering time is calculated by jointly considering the task being processed in the edge server and stored in the buffer. The collaboration target for QoE improvement is formulated using the delay model as follows:(10)$$T_{\hat{j}}=\min\left\{T_{comp,\hat{j}}+T_{comm,\hat{j}}+T_{buf,\hat{j}}\right\}$$
(11)$$s.t.\;T_{\hat{j}}<D_{\hat{j}}$$
(12)$$\sum_{j=1}^{J}R_{j}\leq C_{max,i}$$
(13)$$R_{j}>0$$

The objective function, in Equation (10), aims to minimize the completion time of the task based on the delay model. The constraint, in Equation (11), ensures that the target of collaboration must process tasks within the deadline to improve QoE. The constraint, in Equation (12), ensures that the computing resource needed to process the task does not exceed the edge server’s computing resource capacity. The constraint, in Equation (13), ensures that all the tasks being processed are allocated computing resources from the edge server. At this point, collaboration can be calculated using the greedy algorithm [38].

The details of the greedy-based edge collaboration are described in Algorithm 1. We jointly consider success and completion time to improve QoE. We initialize the collaboration target to the local edge to reduce unnecessary collaboration. In Line 3, the set of edge servers is sorted based on the available computing resources, considering the processing time. In Lines 8–11, we select the collaboration target when the edge server can process the task within the deadline. In Line 8, if the load on the edge server is low, the edge server chooses a collaboration target with sufficient available computing resources. By contrast, in Line 10, even if buffering time occurs, we determine whether the collaboration target can provide sufficient computing resources for the task. Collaboration targets are chosen by expected computing resources and computational threshold. The computational threshold is calculated using Equation (14).
**Algorithm 1** Greedy-based Edge Collaboration  **Input:** Available bandwidth of each link, the computing resource and buffer status of edge computing**Output:** Edge server for cooperation1: **Initialize** the target=i2: **Compute** completion time of task j^ in edge server i3: Sorting the list I′ in decreasing order of computing resource of edge computing4: **For** each edge computing i′∈I′5:  **Compute** computation threshold Thj^6:  **Compute** completion time of task j^7:  **Compute** expected computing resource of task processing8:  **If** Ci′max−Ci′≥Thj^9:     target=i′; **Break**;10:   **Else If** E{Rj}≥Thj^11:   target=i′; Break;12:  Sorting the list I′ in decreasing order of buffer delay13:  **Compute** sustainable delay δd14:  **For** each edge computing i′∈I′15:   **If** Tbuf+Tcomm<δd16:   target=i′; Break;17:   **Else If** Ti′j<Tij18:   target=i′; Break;19:  **Return** target
(14)Thj^=Lj^D−(Tcomm,j^+Tbuf,j^)

The computation threshold represents the computation time needed for a task to be successfully processed. If the edge server does not have available computing resources, the collaboration target is determined based on the computing resources allocated to the task being processed by the edge server. The larger the computing resource allocated, the better the QoE. When the edge server’s computing resources are insufficient and the task cannot be processed within the deadline, we select the collaboration target to reduce the completion time of the task in Lines 13–18. A large number of tasks affects the load on the network and edge servers. Accordingly, in Line 15, we select a collaboration target by considering the transmission time and buffering time. The sustainable delay for selecting a collaboration target is calculated as follows:(15)$$\delta_{d}=D\left(1+\sqrt{\sum_{k=1}^{N}{\left(T_{k}-E\left\{T^{N}\right\}\right)}^{2}}\right)$$

N denotes the number of tasks for calculating the average completion time and is calculated using Equation (16).
(16)$$N=\frac{C_{i}^{max}\cdot D}{L_{j}}$$

If the sustainable delay is not satisfied, the collaboration target is selected according to the predicted completion time by the delay model. The edge collaboration algorithm selects the edge server to which the device is connected when the collaboration target is not found.

### 3.3. Computing Resource Allocation

Tasks are assigned to the edge server according to the collaboration decision and the selection of the collaboration target. The edge server processes the tasks assigned to it and determines the number of tasks to process according to the number of tasks in the buffer. The computing resources allocated to each task are decided by the number of tasks to be processed, which is calculated as in Equation (17).
(17)$$\zeta =\max\{k|\frac{J^{\prime }}{k}<\frac{J^{\prime }}{\beta }\}$$
(18)s.t. k≥1

β denotes the trade-off parameter between the processing and buffering times. When the number of tasks being processed by the edge server is small, the processing time is short; however, the buffering time increases due to the large amount of computing resources allocated to the task. By contrast, when the number of tasks being processed by the edge server is large, the amount of computing resources allocated to the tasks decreases, increasing the completion time. After calculating the number of tasks to be processed, the computing resources allocated to each task are calculated using Equation (19).
(19)$$\zeta =\max\{k|\frac{J^{\prime }}{k}<\frac{J^{\prime }}{\beta }\}$$

## 4. Performance Evaluation

### 4.1. Simulation Setup

To evaluate the performance of the proposed scheme, EdgeCloudSim [39], an extension of CloudSim [40], was used. Figure 4 shows the network topology of edge servers in our simulation. The network between edge servers was configured in a graph structure. Each device has a task creation module, and the task is created according to Poisson distribution. The number of edge servers is set to 10, the computing resources of the edge server is set to 40,000 MIPS, and the bandwidth between the edge servers is set to 100 Mbps, Th is set to 0.7, and β is set to 2. In the experiment, the number of devices connected to each edge server was set randomly. To evaluate the performance of the proposed scheme, we compared it with the *Edge-only*, *Random*, *LCDA*, and *HOM* collaboration schemes. Table 1 shows task parameters in the simulation. Task 1 represents medium sensitivity and small size tasks. Task 2 represents high sensitivity and medium size tasks that requires more computing resources than other tasks for achieving QoE. Task 3 represents low sensitivity and large size tasks that requires fewer resources for QoE compared with the other two task categories.

### 4.2. QoE Performance for Task 1

Figure 5 shows the QoE of collaboration schemes for Task 1. *Edge-only* processes tasks on the edge server to which the device is connected. When the number of devices is small, *Edge-only* achieves low completion, and no communication overhead. However, *Edge-only* has a high completion time and low success rate when the number of devices is large because it does not utilize the computation resources of other edge servers. Therefore, as the number of tasks allocated to an edge server increases, the computing resources allocated to the tasks decrease. Therefore, *Edge-only* has low QoE due to high completion time and low success rates.

*Random* selects collaboration targets regardless of the computing and communication resources. Therefore, it has a high completion time due to unnecessary collaboration when the number of devices is small. The unnecessary collaborations increase the communication time and reduce the QoE compared with other collaboration schemes. However, it has a higher QoE than *Edge-only* with a high success rate (5% better than *Edge-only*) and low completion time (39% less than *Edge-only*) because it uses the computation resources of other edge servers as the number of devices increases.

*LCDA* determines the collaboration target by selecting the edge server’s computing resources upon receiving a task. The *LCDA* selects the lowest loaded edge server for task processing. When the load on the edge server is low, the *LCDA* selects a local edge server as a collaboration target, resulting in low communication time. Nevertheless, *LCDA* shows a higher completion time (21% more) than *Edge-only* due to network overhead. *LCDA* has a low completion time by collaborating with an edge server that is efficient for task processing when the number of devices is large. However, *LCDA* shows a higher communication time than other schemes because the tasks are sent to the edge server according to the availability of the computing resources. Because of this high communication time, *LCDA* shows the lowest success rate (22.8% less than *Edge-only*) when the number of devices is 800.

*HOM* ranks the edge server based on the path. The collaboration target is determined by the rank of edge server. Therefore, *HOM* shows a low completion time (6.2% less than *Random*) when the number of devices is 1000 because *HOM* achieves a lower communication time than other schemes. However, *HOM* selects the collaboration target regardless of the load of the edge server. *HOM* shows an 11% higher processing time and 2% lower success rate than *LCDA* because it offloads the work to high-load collaboration targets.

The proposed scheme determines collaboration based on the available computing resources of the edge servers. After determining the collaboration, the collaboration target is selected according to computation, communication, and buffering time. Therefore, even if the number of devices is small, the proposed scheme achieves a similar performance as the *Edge-only* by reducing the number of collaborations. The proposed scheme guarantees the allotment of the required computing resources for a task even when the load at all edge servers increases. Thus, the proposed scheme has improved QoE, lower completion time (18% less than *LCDA*), and a higher success rate (9.3% more than *HOM*) than the existing collaboration schemes.

### 4.3. QoE Performance for Various Tasks

In this experiment, three types of tasks with different sizes and deadlines are used. To evaluate the proposed scheme, Tasks 1, 2, and 3 are randomly selected with equal probability. Figure 6 shows the QoE of the collaboration scheme for various tasks. *Edge-only* and *Random* are used to process the task regardless of its characteristics. Therefore, efficiency is low regardless of the availability of sufficient computing resources to satisfy the task requirements.

First, *LCDA* classifies the task into latency-sensitive and latency-tolerant according to the task deadline. An appropriate collaboration target is then selected based on the classified tasks. If the task is latency-sensitive, *LCDA* chooses a least-loaded edge server to reduce the computation time. Conversely, if the task is latency-tolerant, *LCDA* selects the edge server with a small number of latency-sensitive tasks processed by the edge server. Because *LCDA* processes differently depending on the task characteristics, a higher success rate (4.4% more than *Random*) and lower completion time (21% less than *Random*) than existing schemes can be achieved.

*HOM* finds the shortest path based on the network bandwidth. If the number of tasks transmitted to the edge server is large, it is determined as a low rank. By ranking according to the network path and network load, *HOM* shows a high success rate (4% more than *Random*) and low task completion time (28% less than *Random*) when the number of devices is 1000. In particular, the processing time is similar to that of *LCDA*; however, it achieves the lowest communication time compared with the other schemes.

The proposed scheme predicts the completion time of a task according to the deadline and size to efficiently collaborate with the edge servers. Therefore, the processing time of the proposed scheme is 2.4% less than *HOM*, and the communication time is 81% lower than *LCDA*. In addition, the proposed scheme achieves an 11.2% higher success rate than *HOM* because the scheme guarantees a high QoE through the optimal allocation of the computing resources.

Figure 7 and Figure 8 show the success rate and completion time of Tasks 1, 2, and 3. *Edge-only* provides sufficient computing resources when the number of devices is small, ensuring that QoE is high for all tasks. However, as the number of devices increases, QoE is degraded. By contrast, *Random* does not consider the task characteristics as *Edge-only*. Therefore, it has a high completion time and low success rate of tasks and requires many computing resources for high QoE in Task 2.

*LCDA* collaborates with other edge servers based on the classified tasks and selects an edge server that can allocate sufficient computing resources for latency-sensitive tasks, such as Task 2. *LCDA* shows 64% lower than *Random*. By contrast, *LCDA* does not efficiently collaborate on latency-tolerant tasks, such as Task 1 and Task 3; hence, the completion time increases as the number of devices increases. *LCDA* shows a low success rate due to the decrease in computing resources as the number of tasks assigned to the edge server increases.

*HOM* calculates the shortest path according to the network bandwidth and task size. This scheme effectively reduces the communication time, resulting in low completion times for all tasks. However, *HOM* cannot determine optimal collaboration according to the task deadline. For this reason, *HOM* shows lower success rates (4.3% in Task 2 and 5.4% in Task 3) than *LCDA*.

The proposed scheme predicts the completion time according to the characteristics of the task and selects the optimum collaboration target that reduces the completion time for each task. In particular, the proposed scheme allocates computing resources to ensure high success rate of each task. When the number of devices is 1000, the completion times of Task 1 and Task 2 using the proposed scheme are 45% and 26% higher than that of the *LCDA*. The reason is that the proposed scheme allocates computing resources regardless of task requirements. Nevertheless, the proposed scheme improves QoE by achieving a 61% low completion time for Task 3.

## 5. Conclusions

In this paper, we propose a resource prediction-based edge collaboration for improving QoE. The proposed scheme probabilistically determines the optimal collaboration target to improve QoE between servers based on predicting computing resources when the edge server receives a task. According to the delay model, the collaboration target is determined using a greedy algorithm based on the task completion time. The amount of computing resources allocated to a task is determined based on the processing and buffering times of the task. Experimental results show that the proposed scheme achieves a low completion time and high success rate according to computing resource allocation and greedy-based edge collaboration. Consequently, the proposed scheme outperforms the QoE of the existing edge collaboration schemes for various tasks.

Future work includes resource allocation based on the characteristic of tasks. In addition, the proposed scheme will be evaluated based on the performance of other parameters and be implemented in real-world edge collaboration systems to investigate its actual applicability and performance.

## Figures and Tables

**Figure 1 sensors-21-08500-f001:**
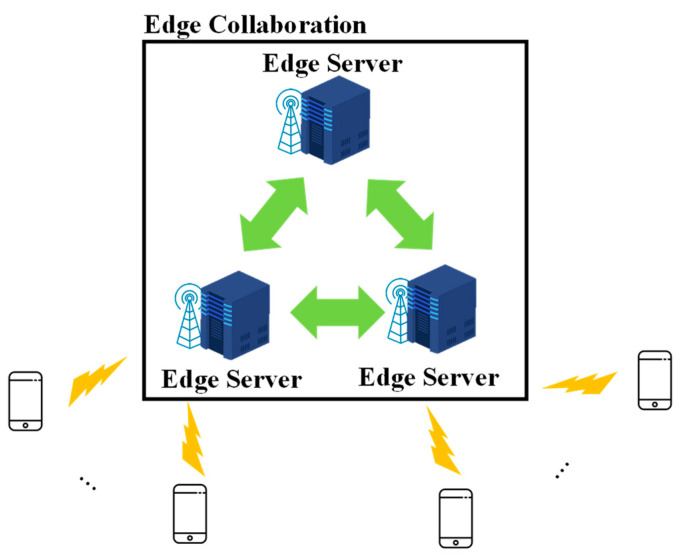
Edge collaborative network environment.

**Figure 4 sensors-21-08500-f004:**
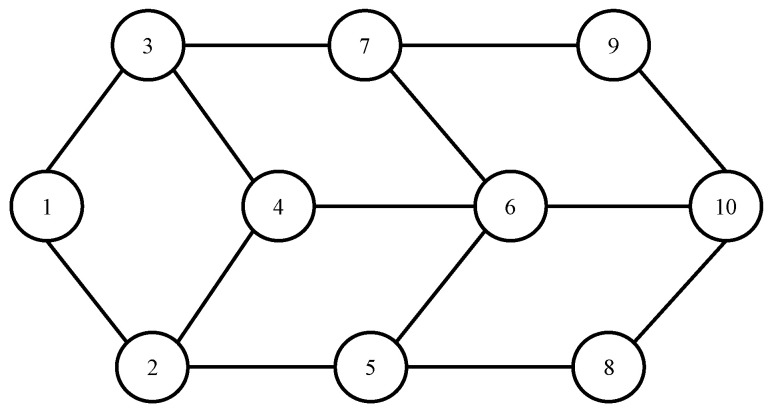
Network topology of edge servers in our simulation.

**Figure 5 sensors-21-08500-f005:**
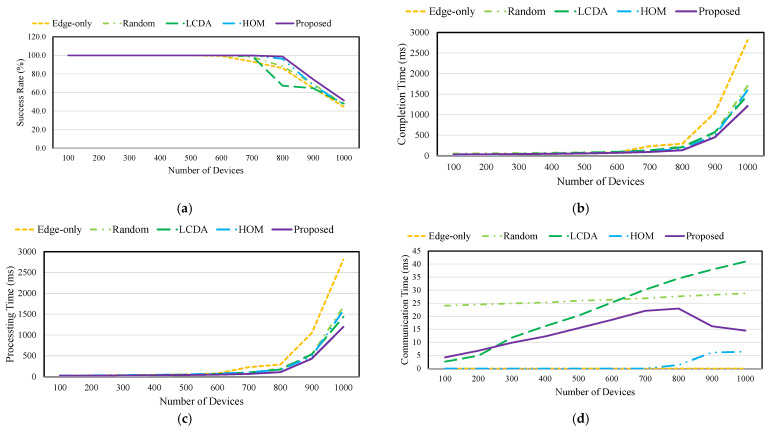
QoE of collaboration schemes for Task 1: (**a**) Success rate; (**b**) Completion time; (**c**) Processing time; (**d**) Communication time.

**Figure 6 sensors-21-08500-f006:**
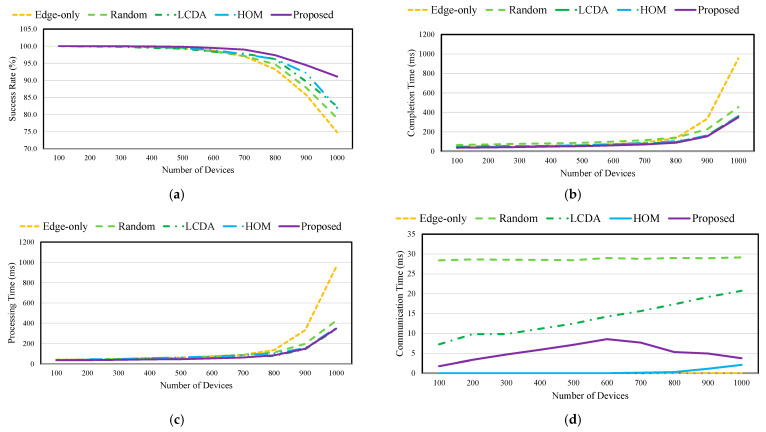
QoE of collaboration schemes for various tasks: (**a**) Success rate; (**b**) Completion time; (**c**) Processing time; (**d**) Communication time.

**Figure 7 sensors-21-08500-f007:**
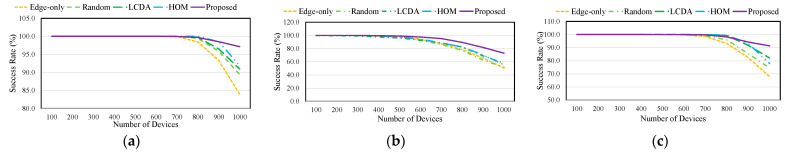
Success rate of Task 1, Task 2, and Task 3: (**a**) Task 1; (**b**) Task 2; (**c**) Task 3.

**Figure 8 sensors-21-08500-f008:**
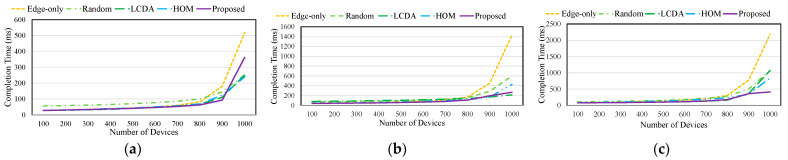
Completion time of Task 1, Task 2, and Task 3: (**a**) Task 1; (**b**) Task 2; (**c**) Task 3.

**Table 1 sensors-21-08500-t001:** Task parameters in the simulation.

	Task 1	Task 2	Task 3
Task Length (MIPS)	1000	1500	3000
Task Deadline (ms)	500	200	800

## Data Availability

Not applicable.

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
