# Peer review of "Resource Prediction-Based Edge Collaboration Scheme for Improving QoE"

_sensors, 2021, doi:10.3390/s21248500_

Round 1
Reviewer 1 Report
This paper propose a resource prediction-based edge collaboration scheme for improving QoE for MEC-based computing services. Particularly, this paper presents the system models of the edge collaborative network and decompose the time delay of services into three parts: processing, communication, and buffering time, which together represent the QoE of services. Then, the edge collaboration can be seen as how the edge server decides to either process the receiving task locally or offload it to a collaborative node via predicting. This collaboration problem is further modelled into an optimization problem of minimizing the total time delay. Next, the authors propose a greedy-based algorithm to the optimization problem and properly allocate the computing resources. Finally, with extensive simulations, the authors validate the effectiveness of the proposed algorithms, in comparison with some baselines.
Overall, the idea of this paper is easy to follow and seems to be convincible. However, the organization and presentation of this paper should be significantly improved to enhance the readability. Besides, I have some concerns summarized as follows.
- How does this work advance the existing work in the field? What are the differences and novelties from the related work.
- As described in Section 2, there are extensive existing work on edge collaboration. What are the chanllenges and contributions of this work This is not clear in the paper.
- The complexity of Algorithm 1 should better be analyzed.
- The authors should first present the basic idea of the algorithm from a high-level in Section 3. This would help to better understand the algorithm.
- The presenation should be significantly improved to enhance the readability, especially Section 3. The notations and figures should be described at the first time they appear. One typical example is that “XXX the threshold (C_i^max Th)” where the notations are given without any explanation until further reading two or three paragraphs.
- Also, Figure 2 is presented without any explanation of details.
- The indices of i and j in E_i,j in Line 170 stands for what? this is not clear.
- There are also many typos and grammatical issues in the paper. I would recommend the paper should be polished by a native English proofreading service. e.g., trade-ff in Line 281, Th is et to 0.7 in Line 295.
Author Response
Reviewer Comment 1:
How does this work advance the existing work in the field? What are the differences and novelties from the related work
Response:
Thank you for your comment. Our scheme improves QoE (Quality of Experience) in terms of task completion time and task success rate. For achieving this goal, we designed the offloading of the task by considering processing and transmission time as well as buffering time. Also, for reducing the task completion time, we propose the computing resource allocation algorithm. We revised the manuscript for clarification.
In the revised version of the paper:
Existing schemes consider processing and transmission time in order to task com-pletion time and process the task within the deadline. In this paper, we consider the pro-cessing and transmission time as well as buffering time. In addition, we propose a scheme for allocating computing resources to process tasks, which reduces task completion time and processes the tasks within the deadline.
Reviewer Comment 2:
As described in Section 2, there are extensive existing work on edge collaboration. What are the challenges and contributions of this work? This is not clear in the paper.
Response:
Thank you for your comment. We updated the manuscript by adding the more description on challenges and contributions of existing work.
In the revised version of the paper:
CACTA [22] considers the uncertainty and stochastic nature of the edge and formulates an optimization problem by modeling the computation capacities and costs of edge nodes using the Autoregressive Integrated Moving Average (ARIMA) model. This scheme im-proves the transmission efficiency of tasks and reduces the task completion time.
By classified tasks, LCDA reduces task completion time and increases task success rates. When an edge server is overloaded, its performance tends to be low, which is evident by a high transmission time. H. Zhao et al. [25] selected edge servers or cloud centers using computation resources and task caching for improving processing and transmission effi-ciency. In contrast, HOM [26] proposed a heuristic method for communication delay minimization. This scheme uses a tree to find the shortest path to the destination edge server and set up a multi-hop network for performance evaluation. However, processing time is high because the load of the collaboration target is not considered.
Reviewer Comment 3:
The complexity of Algorithm 1 should better be analyzed.
Response:
Complexity of algorithm is not main considered in the paper. The complexity of the greedy-based algorithm is shown in reference below.
- Toma and J. Cheng, “Computation Offloading for Fame-Based Real-Time Tasks with Resource Reservation Servers,” Euromicro Conference on Real-Time Systems, Sep. 2013.
- You, K. Huang, H. Chae, and B. Kim, “Energy-Efficient Resource Allocation for Mobile-Edge Computation Offloading,” IEEE Transactions on Wireless Communications, Vol. 16, No. 3, pp. 1397-1411, Dec. 2016.
We added more detail explanations on algorithm 1 in the revised manuscript.
In the revised version of the paper:
In Line 3, the set of edge servers are sorted based on the available computing resources, considering the processing time. In Lines 8-11, we select the collaboration target when the edge server can process the task within the deadline. In Line 8, if the load on the edge server is low, the edge server chooses a collaboration target with sufficient available com-puting resources. By contrast, in Line 10, even if buffering time occurs, we determine whether the collaboration target can provide sufficient computing resources for the task. Collaboration targets are chosen by expected computing resources and computational threshold. The computational threshold is calculated using Eq. (14).
Reviewer Comment 4:
The authors should first present the basic idea of the algorithm from a high-level in Section 3. This would help to better understand the algorithm.
Response:
We describe the basic idea of the algorithm in Section 3. We updated the manuscript by adding detailed explanations to make it easier to understand the operation of the proposed scheme.
In the revised version of the paper:
In this paper, we propose an edge collaboration and computing resource allocation algorithm for improving the QoE in terms of task completion time and task success rate. The edge collaboration is designed by task completion time and a greedy algorithm, and the computing resource allocation is designed by processing and buffering time. In this section, we first introduce the edge collaborative network framework. Second, we focus on the decision-making process in the framework. Finally, the computing resource allocation scheme is described.
Reviewer Comment 5:
The presentation should be significantly improved to enhance the readability, especially Section 3. The notations and figures should be described at the first time they appear. One typical example is that “XXX the threshold (C_i^max Th)” where the notations are given without any explanation until further reading two or three paragraphs.
Response:
We have thoroughly checked and revised our paper for a better presentation. We have asked the professional proofreading and correction service provided by native English editors for further improvement. We have also made the modifications based on the comments by the reviewers.
In the revised version of the paper:
denotes the computing resource usage, denotes the threshold of computing resource usage, and denotes computing resource capacity of edge server.
denotes a set of information about task being processed in the edge server . denotes the index of edge server and denotes the index of task being processed in the edge server .
Reviewer Comment 6:
Also, Figure 2 is presented without any explanation of details.
Response:
We descried the figure 2 in original manuscript.
Fig. 2 shows the operation of the proposed scheme. The computing resource usage of the edge server is first predicted upon receiving the task is received form the devices. Collaboration is determined probabilistically based on the predicted computing resource usage and threshold. According to the collaboration decision, a delay model selects the collaboration target to improve the task success rate uses. The tasks are then assigned to the edge server according to the selected collaboration target. If the edge server has available computing resources, the assigned tasks are processed immediately, else, the task is stored in the edge server’s buffer. The edge server allocates computing resources based on the trade-off between computation and buffering times.
To clarify the explanation of figure 2, we updated further the manuscript.
In the revised version of the paper:
Figure 2 shows the operation of the proposed scheme. denotes the computing resource usage, denotes the threshold of computing resource usage, and denotes the computing resource capacity of the edge server. The task created in the device is transmitted to the connected edge server using the wireless network. The edge server immediately decides offloading whenever it receives a task from the device. The computing resource usage of the edge server is predicated upon receiving the task from the devices. If the predicted computing resource usage is lower than the threshold, the task is processed in a local edge server; otherwise, a collaboration is determined probabilistically based on the predicted computing resource usage and threshold. We first model the completion time of the task and formulate an objective function to determine the optimal collaboration. To solve the complex problem of the objective function, a collaboration target is selected using a greedy algorithm. The tasks are then assigned to the edge server according to the selected collaboration target. If the edge server has available computing resources, the assigned tasks are processed immediately; otherwise, the task is stored in the edge server’s buffer. The edge server counts the number of offloaded tasks to be processed. The computing resource allocation is calculated by a trade-off between processing and buffering time.
Reviewer Comment 7:
The indices of i and j in E_i,j in Line 170 stands for what? this is not clear.
Response:
We have made the modifications based on the comments by the reviewer.
In the revised version of the paper:
The edge server processes the task received from the device and is aware of the task being processed . denotes a set of information about task being processed in the edge server . denotes the index of the edge server.
Reviewer Comment 8:
There are also many typos and grammatical issues in the paper. I would recommend the paper should be polished by a native English proofreading service. e.g., trade-ff in Line 281, Th is et to 0.7 in Line 295.
Response:
We corrected our manuscript based on the reviewer’s comments. We have also thoroughly checked and revised our paper for a better expression. In addition, we have also asked the professional proofreading and correction service provided by native English editors for further improvement.

Reviewer 2 Report
The General writing and organization of the paper is good, timely required work. The methodology is promising and clearly represented.
Some recent related work 2020 and 2021 are good to be added to this work.
The experimental results are required to be compared with more recent studies.
Other parameters can be evaluated to enrich the work.
General proofreading is required to remove typos and grammatical issues
Author Response
Reviewer Comment 1:
Some recent related work 2020 and 2021 are good to be added to this work.
Response:
Thank you for your comment. As the reviewer suggested, recently published papers have been added to References (Ref. No. 34, 35) and discussed in Section 2.
In the revised version of the paper:
- Gong [34] established the optimal communication scheduling and computation allocation. Based on communication scheduling and computation allocation, Gong found optimal computation nodes to reduce the task completion time. D3PG [35] partially offloads the task to an edge server. To minimize the task completion time, D3PG decides the offloading size of the task and network resource to the task based on reinforcement learning, which is designed based on a dueling and double network architecture to reduce the con-vergence time.
Gong. X. Delay-Optimal Distributed Edge Computing in Wireless Edge Networks. IEEE INFOCOM, 2020, 2629-2638.
Lu H.; He X.; Du M.; Ruan X.; Sun Y. Wang K. Edge QoE: Computation Offloading with Deep Reinforcement Learning for Internet of Things. IEEE Internet of Things J. 2020, 7, 9255-9265.
Reviewer Comment 2:
The experimental results are required to be compared with more recent studies.
Response:
Thank you for your opinion. We added it to the experiment as comparison scheme that selects collaboration target based on network. We updated the manuscript by adding new collaboration scheme to the experimental results.
In the revised version of the paper:
To evaluate the performance of the proposed scheme, we compared it with the Edge-only, Random, LCDA, and HOM collaboration scheme.
HOM ranks the edge server based on the path. The collaboration target is determined by rank of edge server. Therefore, HOM shows a low completion time (6.2% less than Random) when the number of devices is 1000. This is because HOM achieves a lower communication time than other schemes. However, HOM selects the collaboration target regardless of the load of the edge server. HOM shows a 11% higher processing time and 2% low success rate than LCDA because it offloads the work to high-load collaboration targets.
HOM finds the shortest path based on the network bandwidth. If the number of tasks transmitted to the edge server is large, it is determined as a low rank. By ranking according to the network path and network load, HOM shows high success rate (4% above than Random) and low task completion time (28% than Random) when the number of devices is 1000. In particular, the processing time is similar to that of LCDA; however it achieves the lowest communication time compared with other schemes.
HOM calculates the shortest path according to the network bandwidth and task size. This scheme effectively reduces the communication time, resulting in low completion times for all tasks. However, HOM cannot determine optimal collaboration according to the task deadline. For this reason, HOM shows lower success rates (4.3% in Task 2 and 5.4% in Task 3) than LCDA.
Reviewer Comment 3:
Other parameters can be evaluated to enrich the work.
Response:
We judged that success rate, completion time, processing time, and communication time are important inters of edge collaboration from various QoE perspectives. We will consider additional parameters to evaluate the performance of proposed scheme. We revised the manuscript by adding the additional parameters to evaluate the performance in Section 5.
In the revised version of the paper:
Future work includes resource allocation based on the characteristic of tasks. In addition, the proposed scheme will be evaluated based on the performance of other parameters and be implemented in real-world edge collaboration systems to investigate its actual applicability and performance.
Reviewer Comment 4:
General proofreading is required to remove typos and grammatical issues.
Response:
We have thoroughly checked and revised our paper for a better expression. We have also asked the professional proofreading and correction service provided by native English editors for further improvement.
Please see the attachment

Reviewer 3 Report
The manuscript proposes a novel scheme for the scheduling of computational task on multiple Edge Servers. The scheme aims to solve the QoE reduction resulting from the overloading of Servers, when the number of devices requesting computational resources rises. The overall methodology seems sound, and it is supported by adequate simulation results.
However the paper lacks a proper comparison with existing scheduling schemes, and the introduction only states the existence of this QoE problem, without providing any reference for the reported claims. While the simulation takes in consideration other possible schemes, these seem to be too general and not referring to any real implementation. The related works section is limited to a list of existing approaches, some of which are also quite old, but without a proper evaluation and critic.
The paper also requires some proofreading, to correct some typos and misspellings, also in the references.
Author Response
Reviewer Comment 1:
the paper lacks a proper comparison with existing scheduling schemes, and the introduction only states the existence of this QoE problem, without providing any reference for the reported claims.
Response:
Thank you for your comment. As the reviewer suggested, recently published papers have been added to References (Ref. No. 6, 7)
In the revised version of the paper:
The existing edge collaboration scheme determines the collaboration target without considering the computing resource or network resource [6,7].
Wang J.; Pan J.; Esposito F.; Calvam P.; Yang Z. Edge Cloud Offloading Algorithms: Issues, Methods, and Perspectives. ACM Compu. Sury. 2019, 52.
Jiang C.; Cheng X.; Gao H.; Zhou X.; Wan J.; Toward Computation Offloading in Edge Computing: A Survey. IEEE Access. 2019, 7, 131543-131558.
Reviewer Comment 2:
While the simulation takes in consideration other possible schemes, these seem to be too general and not referring to any real implementation.
Response:
Thank you for your opinion. We added it to the experiment as comparison scheme that selects collaboration target based on network. We updated the manuscript by adding new collaboration scheme to the experimental results.
In the revised version of the paper:
To evaluate the performance of the proposed scheme, we compared it with the Edge-only, Random, LCDA, and HOM collaboration scheme.
HOM ranks the edge server based on the path. The collaboration target is determined by rank of edge server. Therefore, HOM shows a low completion time (6.2% less than Random) when the number of devices is 1000. This is because HOM achieves a lower communication time than other schemes. However, HOM selects the collaboration target regardless of the load of the edge server. HOM shows a 11% higher processing time and 2% low success rate than LCDA because it offloads the work to high-load collaboration targets.
HOM finds the shortest path based on the network bandwidth. If the number of tasks transmitted to the edge server is large, it is determined as a low rank. By ranking according to the network path and network load, HOM shows high success rate (4% above than Random) and low task completion time (28% than Random) when the number of devices is 1000. In particular, the processing time is similar to that of LCDA; however it achieves the lowest communication time compared with other schemes.
HOM calculates the shortest path according to the network bandwidth and task size. This scheme effectively reduces the communication time, resulting in low completion times for all tasks. However, HOM cannot determine optimal collaboration according to the task deadline. For this reason, HOM shows lower success rates (4.3% in Task 2 and 5.4% in Task 3) than LCDA.
Reviewer Comment 3:
The related works section is limited to a list of existing approaches, some of which are also quite old, but without a proper evaluation and critic.
Response:
We agree that some existing schemes are quite old references and there are no evaluations or criticisms of them. However, very old references are needed to explain the background to the offloading schemes. The manuscript has been updated to reflect the reviewers' opinions.
In the revised version of the paper:
By classified tasks, LCDA reduces task completion time and increases task success rates. When an edge server is overloaded, its performance tends to be low, which is evident by a high transmission time. H. Zhao et al. [25] selected edge servers or cloud centers using computation resources and task caching for improving processing and transmission efficiency. In contrast, HOM [26] proposed a heuristic method for communication delay minimization. This scheme uses a tree to find the shortest path to the destination edge server and set up a multi-hop network for performance evaluation. However, processing time is high because the load of the collaboration target is not considered.
Reviewer Comment 4:
The paper also requires some proofreading, to correct some typos and misspellings, also in the references.
Response:
We have thoroughly checked and revised our paper for a better expression. We have also asked the professional proofreading and correction service provided by native English editors for further improvement.
Please see the attachment

Reviewer 4 Report
This paper is well described in the sense that the main claim to improve the system performance and the developed algorithm to realize the objective are clearly described, and the simulation showed that the intended improvement is obtained. The paper will provide some benefits to readers who have interest in this field.
To increase the value of the paper, following points are strongly recommended for revision.
- Section 3, the most important part of the paper, needs the sentence brush. Some editorial errors are seen. It is preferred that some sentences are modified to make the readers easily understand your intention. One simple example is that the use of “processed” sometimes seems to mean “being processed” or “under process”.
- Please check the equation 4. It seems that Eij is the assembly of tasks whose components are the set of remaining task size and assigned computational power of each task. What is the value of Eij. And what is D?
- In your design, is it assumed that the different task requests arrive almost the same time to the different nodes? If not, it is better to indicate this as one of the major assumptions of the system behavior.
- Section 4, another important part of the paper needs more description on the simulation conditions.
- Are the devices equally distributed to each node?
- Imaging that each node receives task request from devices connected to it, how the arrival is designed? It is written that the task creation is done base on Poisson distribution, but whether each node has its own task creation scheduler or total system has one task creation scheduler is not clear. In the latter case, it is necessary to describe how the created event is assigned to which node.
- Location of Fig.5 is better changed to sit within Section 4.2, and Fig.6 within the section4.3
- Line 296, what is I?
Author Response
Reviewer Comment 1:
Section 3, the most important part of the paper, needs the sentence brush. Some editorial errors are seen. It is preferred that some sentences are modified to make the readers easily understand your intention. One simple example is that the use of “processed” sometimes seems to mean “being processed” or “under process”.
Response:
Thank you for your comment. We have thoroughly checked and revised our paper for a better expression. We have also asked the professional proofreading and correction service provided by native English editors for further improvement.
In the revised version of the paper:
Therefore, the proposed scheme reduces the collaboration probability based on the state of the task being processed in the edge server.
If the edge server does not have available computing resources, the task is processed after the task being processed on the edge server is completed. Therefore, the computation time of the task is calculated based on the computing resources allocated to the task being processed by the edge server.
Reviewer Comment 2:
Please check the equation 4. It seems that Eij is the assembly of tasks whose components are the set of remaining task size and assigned computational power of each task. What is the value of Eij. And what is D?
Response:
Thank you for your comment. and its components are described in lines 170 through 173. denotes set of tasks being processed in the edge server . , which is a component of , represents remaining size until completion of the task processed by the edge server and allocated computing resources for the task, respectively. denotes the deadline of task . We revised the manuscript for clarification.
In the revised version of the paper:
(1)
denotes the size of the task and denotes the processing deadline of the task .
denotes set of information about task being processed in the edge server.
|
|
(3) |
|
(4) |
Reviewer Comment 3:
In your design, is it assumed that the different task requests arrive almost the same time to the different nodes? If not, it is better to indicate this as one of the major assumptions of the system behavior.
Response:
Tasks are randomly generated on the devices. Accordingly, task requests can be made at the same time. In our design, each edge server handles task requests immediately. We updated the manuscript by adding more detail description on the assumption with different task requests.
In the revised version of the paper:
In this paper, we make three major assumptions. First, the device does not have sufficient computing resources to process the task. Therefore, whenever a task is created, the device transmits the task and task information to a connected edge server. Second, the edge server knows the bandwidth of the network for task transmission. Finally, multiple tasks can be created concurrently using different devices.
Reviewer Comment 4:
Are the devices equally distributed to each node?
Response:
We did not set to the devices equally distributed to each node. The devices are randomly connected to edge server. We updated the manuscript for clarification.
In the revised version of the paper:
In the experiment, the number of devices connected to each edge server was set randomly.
Reviewer Comment 5:
Imaging that each node receives task request from devices connected to it, how the arrival is designed? It is written that the task creation is done base on Poisson distribution, but whether each node has its own task creation scheduler or total system has one task creation scheduler is not clear. In the latter case, it is necessary to describe how the created event is assigned to which node
Response:
Thank you for your comment. The device is connected to edge server via wireless network. When task is created from the device, the task is transmitted to connected edge server. The edge server immediately decides offloading the tasks. Also, each device has task creation module. We updated the manuscript for clarification.
In the revised version of the paper:
The task created in the device is transmitted to the connected edge server using the wireless network. The edge server immediately decides offloading whenever it receives task form the device.
Each device has a task creation module, and the task is created according to Poisson distribution.
Reviewer Comment 6:
Location of Fig.5 is better changed to sit within Section 4.2, and Fig.6 within the section4.3
Response:
We have made the modifications based on the comments by the reviewers.
Reviewer Comment 7:
Line 296, what is I?
Response:
We are sorry for making a mistake in the manuscript. We revised the manuscripts for clarification.
In the revised version of the paper:
To evaluate the performance of the proposed scheme, we compared it with the Edge-only, Random, LCDA, and HOM collaboration scheme.
Please see the attachment

Round 2
Reviewer 1 Report
Thanks for the revision. Overall, the current version have well addressed my previous concerns. Considering the topic about resource utilization for edge collaboration, I noticed that the following papers might be discussed as the related work. [1] When Edge Meets Learning: Adaptive Control for Resource-Constrained Distributed Machine Learning, IEEE INFOCOM 2018. [2] OL4EL: Online Learning for Edge-Cloud Collaborative Learning on Heterogeneous Edges with Resource Constraints, IEEE Communications Magazine 2020.Author Response
RESPONSE TO THE REVIEWER
REVIEWER
Reviewer Comment 1:
Considering the topic about resource utilization for edge collaboration, I noticed that the following papers might be discussed as the related work. [1] When Edge Meets Learning: Adaptive Control for Resource-Constrained Distributed Machine Learning, IEEE INFOCOM 2018. [2] OL4EL: Online Learning for Edge-Cloud Collaborative Learning on Heterogeneous Edges with Resource Constraints, IEEE Communications Magazine 2020.
Response:
Thank you for your comment. As the reviewer suggested, we add two papers to Reference (Ref. No. 36, 37) and discussed in Section 2.
In the revised version of the paper:
- Wang et al. [36] analyzed the convergence rate of distributed gradient descent. Based on the analyzed results, they determined the best parameter between local update and global aggregation. OL4EL [37] supports both synchronous and asynchronous learning patterns, and can be used for both supervised and unsupervised learning tasks.
Wang S.; Tuor T.; Salonidis T.; Leung K.; Makaya C.; He T.; Chan K. When Edge Meets Learning: Adaptive Control for Resource_Constrained Distributed Machine Learning. IEEE INFOCOM. 2018.
Han Q.; Yang S.; Ren X.; Zhao C.; Zhang J.; Yang X. OL4EL: Online Learning for Edge-Cloud Collaborative Learning on Heterogeneous Edges with Resource Constraints. IEEE Communications Magazine. 2020, 58, 49-55.
Reviewer 3 Report
The authors have properly addressed the issues pointed out after the first round of review, so I suggest to accept the paper.
Author Response
RESPONSE TO THE REVIEWER
REVIEWER
Reviewer Comment 1:
Considering the topic about resource utilization for edge collaboration, I noticed that the following papers might be discussed as the related work. [1] When Edge Meets Learning: Adaptive Control for Resource-Constrained Distributed Machine Learning, IEEE INFOCOM 2018. [2] OL4EL: Online Learning for Edge-Cloud Collaborative Learning on Heterogeneous Edges with Resource Constraints, IEEE Communications Magazine 2020.
Response:
Thank you for your comment. As the reviewer suggested, we add two papers to Reference (Ref. No. 36, 37) and discussed in Section 2.
In the revised version of the paper:
- Wang et al. [36] analyzed the convergence rate of distributed gradient descent. Based on the analyzed results, they determined the best parameter between local update and global aggregation. OL4EL [37] supports both synchronous and asynchronous learning patterns, and can be used for both supervised and unsupervised learning tasks.
Wang S.; Tuor T.; Salonidis T.; Leung K.; Makaya C.; He T.; Chan K. When Edge Meets Learning: Adaptive Control for Resource_Constrained Distributed Machine Learning. IEEE INFOCOM. 2018.
Han Q.; Yang S.; Ren X.; Zhao C.; Zhang J.; Yang X. OL4EL: Online Learning for Edge-Cloud Collaborative Learning on Heterogeneous Edges with Resource Constraints. IEEE Communications Magazine. 2020, 58, 49-55.